# RiemannGFM: Learning a Graph Foundation Model from Structural Geometry

## Abstract

The foundation model has heralded a new era in artificial intelligence, pretraining a single model to offer cross-domain transferability on different datasets. Graphs are omnipresent non-Euclidean structures, ranging from recommender systems to biochemical structures. Graph neural networks excel at learning graph data, but often lack the generalization capacity. Hence, **graph foundation model** is drawing increasing attention, and recent efforts have been made to leverage Large Language Models, encouraged by the remarkable success of GPT-4. On the one hand, existing studies primarily focus on text-attributed graphs, while a wider range of real graphs do not contain fruitful textual attributes. On the other hand, the sequential graph description tailored for the Large Language Model neglects the structural complexity, which is a predominant characteristic of the graph. Such limitations motivate an important question: **Can we go beyond Large Language Models, and pretrain a universal model to learn the structural knowledge for any graph?** The answer in the language or vision domain is a shared vocabulary. We observe the fact that there also exist shared substructures underlying graph domain, and thereby open a new opportunity of graph foundation model with **structural vocabulary** (by which any graph can be constructed). The key innovation of this paper is the discovery of a simple yet effective structural vocabulary of trees and cycles, and we explore its inherent connection to Riemannian geometry. Herein, we present a universal pretraining model, `RiemannGFM`, with geometric contrastive learning. Concretely, we first construct a novel product bundle to incorporate the diverse geometries of the vocabulary. On this constructed space, we stack Riemannian layers where the structural vocabulary, regardless of specific graph, is learned in Riemannian manifold. This offers the shared structural knowledge for cross-domain transferability, and node encoding is generated in the tangent space for arbitrary input graph. Empirical results show the superiority of `RiemannGFM` on a diversity of real graphs.

## Keywords

Graph Foundation Model, Riemannian Geometry, Pretraining Model.

**ACM Reference Format:**
Anonymous Author(s). 2024. RiemannGFM: Learning a Graph Foundation Model from Structural Geometry. In *Proceedings of ACM The Web Conference (ACM TheWebConf '25)*. ACM, New York, NY, USA, 12 pages. https://doi.org/10.1145/nnnnnnn.nnnnnnn

## 1 Introduction

Designing a foundation model has been a longstanding objective in artificial intelligence that pre-trains a single, universal model on massive data allowing for cross-domain transferability on different datasets. Recently, the Large Language Model (LLM) such as GPT-4 [28] marks a revolutionary advancement of the foundation model in the language realm. In the real world, graphs are also ubiquitous, describing the data from Web applications, social networks, biochemical structures, etc. Unlike word sequences in the language, graphs present distinct, non-Euclidean structures encapsulating the complex intercorrelation among objects, which prevents the direct deployment of LLM. Graph Neural Networks (GNNs) [10, 20, 36, 38] conduct neighborhood aggregation over the graph and achieve state-of-the-art performance on learning graph data. The significant limitation of GNNs is the lack of generalization capacity. GNNs are often designed for specific tasks, and re-training is typically required on different tasks or datasets to maintain expressiveness. Consequently, **Graph Foundation Models** (GFMs) are emerging as an interesting research topic in the graph domain.

Compared to the tremendous success of the foundation model in other domains, GFM is still in the infant stage. Pioneering work [35] designs the graph prompting to unify the downstream tasks, the analogy to language prompt. GCOPE [47] further conducts model training on multi-domain graphs with coordinators in order to improve the generalization capacity to different datasets. Recent efforts have been made to integrate GNN with LLM. For example, LLaGA [5] develops a graph translation technique that reshapes a graph into node sequences, while OFA [22] unifies different graph data by describing nodes and edges with natural language. Also, there are successful practices in specific domains (e.g., knowledge graphs [9, 16] and molecular structures [2]) or specific tasks (e.g., node classification [48]), which are far from the universal GFM.

On the one hand, existing studies primarily focus on the text-attributed graphs. The structural knowledge is coupled with textual attributes (or language description), and the transferability relies on the commonness of text [24]. Consequently, it leads to suboptimal performance on the graphs other than text-attributed ones, as investigated in our Experiment as well. However, there exists a wide range of graphs that do not contain fruitful textual attributes, and may only have structural information. An alternative perspective is to seek the transferability from the structures (e.g., the common substructures), so that such knowledge is applicable to any graph. Surprisingly, it has not yet been touched in the context of GFM.

On the other hand, the sequential graph description, tailored for the language model, tends to fail in capturing the structural complexity, which is a predominant characteristic of graphs. GFMs so far work with the traditional Euclidean space, while recent advances report superior expressiveness of hyperbolic spaces in learning tree-like (hierarchical) graphs [3, 27]. However, modeling the

structural complexity is challenging. For instance, hyperbolic models trained on tree-like graphs cannot be generalized to those of different structures. In addition, graph structures are indeed quite complex, tree-like in some regions and cyclical in others [12]. We also notice the product manifold is leveraged to fine-tune the geometry of given structures [7, 34], which is orthogonal to our focus. In other words, it is still not clear how to connect such geometric expressiveness to GFM.

Motivated by the aforementioned limitations, we raise an important question: **Can we go beyond Large Language Models, and pre-train a universal model to learn the structural knowledge for any graph?** The answer to the foundation model in language or vision domain is a shared vocabulary [4]. In fact, there also exist common substructures underlying the graph domain, and the observation offers us a fresh perspective to build graph foundation models. Accordingly, we introduce the concept of **structural vocabulary** by which any graph can be constructed. The key innovation of this paper is the discovery of a simple yet effective structural vocabulary consisting of substructures of trees and cycles (e.g., node triangles). We explore the inherent connection between the structural vocabulary and **Riemannian geometry**, where hyperbolic space aligns tree structures [10, 32], while hyperspherical space is suitable to cycles [12, 29].

Accordingly, it calls for a representation space to model both local geometry (trees or cycles) and graph structure. To this end, we for the first time introduce the tangent bundle to the graph domain, coupling a Riemannian manifold and its surrounding tangent spaces. We leverage the node coordinate on the manifold to embed the local geometry, while the node encoding in tangent spaces accommodates the information of graph structure. Grounded on the elegant framework of Riemannian geometry, we present a universal pre-training model (`RiemannGFM`) on the product bundle to incorporate the vocabulary of diverse geometries. `RiemannGFM` stacks the universal Riemannian layer, which consists of a vocabulary learning module and a global learning module. In the vocabulary learning module, we focus on embedding the structural vocabulary into Riemannian manifolds, regardless of the specific graph. Specifically, this involves updating the node coordinates of a tree (or cycle) in hyperbolic (or hyperspherical) space. For each substructure, cross-geometry attention is formulated in the manifolds in which we derive a manifold-preserving linear operation. The global learning module is responsible for updating the node encoding. With multiple substructures sampled in the graph, we first perform substructure-level aggregation by the proposed bundle convolution, solving the incompatibility issue over tangent bundle, and then calculate the graph-level node encoding with the geometric midpoint and parallel transport. Finally, we conduct geometric contrastive learning among different views, provided by different geometries, so that `RiemannGFM` is capable of generating informative node encoding for an arbitrary graph, underpinned by the shared structural knowledge learned in Riemannian geometry.

**Contribution Highlights.** Overall, key contributions are three-fold: **A. Foundation Model for Graph Structures.** We explore GFM for a wider range of real graphs, not limited to text-attributed ones, and for the first time study GFM from structural geometry to the our best knowledge. **B. Universal Riemannian Pre-training.** We propose a universal pre-trained model (`RiemannGFM`) on a novel product bundle where the structural vocabulary is learned in Riemannian manifold, offering the shared structural knowledge for cross-domain transferability. **C. Extensive Experiments.** We evaluate the superiority of `RiemannGFM` in cross-domain transfer learning and few-shot learning on a diversity of real graphs.

## 2 Preliminaries

This section reviews the basic concepts of Riemannian Geometry and the model space of Lorentz/Spherical Model, and then formally describes the novel problem of graph foundation model from structural geometry. Important notations are summarized in Appx. A.

***Riemannian Geometry.*** Geometrically, a complex structure is related to a Riemannian manifold, which is a smooth manifold $\mathcal{M}$ endowed with a Riemannian metric $\mathfrak{g}$. Each point $\boldsymbol{x}$ in the manifold is associated with a tangent space $\mathcal{T}_x \mathcal{M}$ where the metric $\mathfrak{g}$ is defined. The mapping between the tangent space and manifold is done via exponential and logarithmic maps, and parallel transport conducts the transformation between two tangent spaces. The geodesic between two points is the curve of the minimal length that connects them on the manifold. The curvature $\kappa_x$ is the geometric quantity measuring the extent of how a surface deviates from being flat at $\boldsymbol{x}$. A manifold is referred to as a **Constant Curvature Space** (CCS) if and only if curvature $\kappa_x$ is equal everywhere, so that the closed-form metric is derived. There exist three types of CCS (a.k.a. isotropic manifold): hyperbolic space $\mathcal{H}$ with negative curvature, hyperspherical space $\mathcal{S}$ with positive curvature, and zero-curvature Euclidean space, a special case of Riemannian geometry.

***Lorentz/Spherical Model.*** Here, we give a unified formalism of hyperbolic and hyperspherical space. Specifically, a $d$-dimensional CCS with constant curvature $\kappa$ ($\kappa \neq 0$) is defined on the smooth manifold of $\mathcal{L}_\kappa^d = \{\boldsymbol{x} = \begin{bmatrix} x_t \\ \boldsymbol{x}_s \end{bmatrix} \in \mathbb{R}^{d+1} | \langle \boldsymbol{x}, \boldsymbol{x} \rangle_\kappa = \frac{1}{\kappa}, x_t > 0, \boldsymbol{x}_s \in \mathbb{R}^d\}$ equipped with the curvature-aware inner product as follows,

$$\langle \boldsymbol{x}, \boldsymbol{y} \rangle_\kappa := \operatorname{sgn}(\kappa) x_t y_t + \boldsymbol{x}_s^\top \boldsymbol{y}_s, \quad \boldsymbol{x}, \boldsymbol{y} \in \mathcal{L}_\kappa^d, \quad (1)$$

where sgn is the sign function and thereby the Riemannian metric at $\boldsymbol{x}$ is induced as $\mathfrak{g}_x = \operatorname{diag}(\operatorname{sgn}(k), 1, \dots, 1)$, a diagonal matrix. The north pole of $\mathcal{L}_\kappa^d$ is given as $\boldsymbol{o} = [\frac{1}{\sqrt{|\kappa|}}, 0, \cdots, 0]^\top$. Closed-form exponential and logarithmic maps exist (detailed in Appendix D). In particular, $\mathcal{L}_\kappa^d$ becomes the Lorentz model of hyperbolic space (a.k.a. hyperboloid model) under negative $\kappa$, and shifts to Spherical model of hyperspherical space when $\kappa > 0$.

***Notations & Problem Formulation.*** A graph $\mathcal{G}$ is defined over node set $\mathcal{V}$ and edge set $\mathcal{E} \subseteq \mathcal{V} \times \mathcal{V}$, and each node is optionally associated with an attribute $\boldsymbol{x}$. Note that, the attribute is not necessarily required given a wide range of non-attributed graphs exist in the real world. Analogy to the foundation model for the language, the graph foundation model aims to pre-train a single, universal model $\Phi$ parameterized by $\Theta$, which is applicable to other graphs to generate informative representations $\boldsymbol{z}$ for downstream tasks (e.g., node-level and edge-level). In particular, the model $\Phi$ offers the **cross-domain transferability** that the parameters $\Theta$ pre-trained on one domain can be utilized on another domain with slight treatments. In this paper, we argue that GFM should also offer the **universality** to any graph (not limited to textual-attributed graphs), and highlight the inherent **structural geometry** which is largely ignored in the previous GFMs.

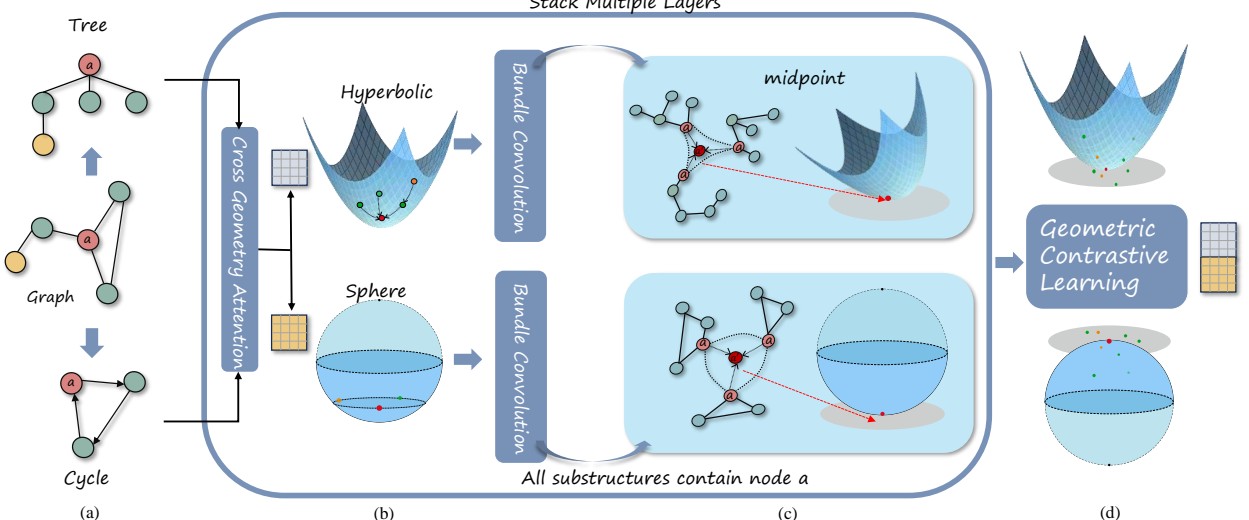

**Figure 1: Overall architecture of the proposed graph foundation model: `RiemannGFM`.**

## 3 RiemannGFM: Learning Structural Vocabulary in Riemannian Geometry

Different from previous GFMs coupling the structural knowledge with textual attributes, we put forward a fresh perspective of studying graph structures, and propose a universal pre-training model named `RiemannGFM`, which is capable of learning the structural knowledge so as to offer the cross-domain transferability among a wider range of real graphs. The key novelty lies in that we discover an effective *structural vocabulary* for any graph structure and explore its connection to *Riemannian Geometry*. At the beginning, we introduce a novel concept of structural vocabulary.

> **DEFINITION 1 (STRUCTURAL VOCABULARY).** *A collection of substructures is said to be a structural vocabulary when they are able to construct an arbitrary graph.*

***Structural Vocabulary and Constant Curvature Spaces (CCS).*** The answer to the foundation model in language or vision domain is a shared vocabulary. For a language model, the text is broken down into smaller units such as words by which the commonness and transferability are encoded [4]. Analogous to word vocabulary of the language, the structural vocabulary considers the shared units in graph domain, . In `RiemannGFM`, we leverage a simple yet effective structural vocabulary, consisting of trees and cycles. An intuitive example is that a tree and cycles with coinciding edges form an arbitrary connected component unless it is exactly a tree. On modeling the vocabulary, we notice the fact that a tree cannot be embedded in Euclidean space with bounded distortion[1], while the embedding distortion is proved to be bounded in low-dimensional hyperbolic spaces [32]. The geometric analogy of cycle is the hyperspherical space, as both cycle and hyperspherical space present the rotational invariant [29]. Therefore, we propose to utilize the constant curvature spaces (i.e., hyperbolic and hyperspherical spaces) to model the substructures of trees and cycles. Note that there exists

no isometric mapping between CCS and Euclidean one, and we discuss the significance of introducing CCS in the Experiment.

***Overall Architecture.*** We design the universal pre-training model (`RiemannGFM`) on the product bundle in light of the diverse substructures in the vocabulary. The overall architecture is illustrated in Fig. 1. According to the structural vocabulary, we first sample the trees and cycles in the graph, as shown in Fig. 1(a). Subsequently, we stack the universal Riemannian layers on the product bundle, consisting of the Vocabulary Learning Module in Fig. 1(b) and Global Learning Module in Fig. 1(c). Without graph augmentation, `RiemannGFM` is pre-trained with the geometric contrastive loss in Fig. 1(d).

### 3.1 Universal Riemannian Layer

In the heart of `RiemannGFM`, we stack the universal Riemannian layer on the product bundle, where Vocabulary Learning Module performs cross-geometry attention to **embed the structural vocabulary into CCSs, regardless of specific graphs, thus offering the shared structural knowledge for cross-domain transferability**. Global Learning Module aligns different substructures with a global view and, accordingly, generates node encodings through the proposed bundle convolution.

*3.1.1 A Layer on the Product Bundle.* We elaborate on a novel representation space for GFM, where the **tangent bundle** is introduced to graph domain for the first time. In Riemannian geometry, a tangent bundle[2] typically consists of (1) a CCS highlighting the local geometry, and (2) the tangent spaces describing the complementary information [29]. In our design, each node $i$ in the bundle is associated with node coordinate and node encoding. Concretely, the coordinate in the manifold $\boldsymbol{p}_i \in \mathcal{M}$ contains the relative position in substructures (i.e., structural vocabulary), while the encoding in tangent space $\boldsymbol{z}_i \in \mathcal{T}_{\boldsymbol{p}_i}\mathcal{M}$ carries the information of global structure (in the graph level). To incorporate the substructures of different geometries, we construct a **product bundle** as

$$\mathcal{P}^{d_P} = \left( \mathcal{H}_{\kappa_H}^{d_H} \otimes \mathcal{T}\mathcal{H}_{\kappa_H}^{d_H} \right) \otimes \left( \mathcal{S}_{\kappa_S}^{d_S} \otimes \mathcal{T}\mathcal{S}_{\kappa_S}^{d_S} \right), d_P = 2d_H + 2d_S, \quad (2)$$

---

[1]Embedding distortion is measured by $\frac{1}{|\mathcal{V}|^2}\sum_{ij}\left|\frac{d_G(v_i,v_j)}{d(\mathbf{x}_i,\mathbf{x}_j)}-1\right|$, where each node $v_i \in \mathcal{V}$ is embedded as $\mathbf{x}_i$ in representation space. $d_G$ and $d$ denote the distance in the graph and the space, respectively.

[2]A tangent bundle is defined as a smooth manifold $\mathcal{M}$ attached with a disjoint union of tangent spaces surrounding it $\mathcal{T}\mathcal{M} = \bigsqcup_{\boldsymbol{x}\in\mathcal{M}} \mathcal{T}_{\boldsymbol{x}}\mathcal{M}$.

where $\otimes$ denotes Cartesian product, $d_{(\cdot)}$ and $\kappa_{(\cdot)}$ are the dimension and curvature, respectively. For each node in this product, we have $\boldsymbol{x}_i = [\boldsymbol{p}_i^H || z_i^H || \boldsymbol{p}_i^S || z_i^H] \in \mathcal{P}^{d_P}$, where $||$ is vector concatenation, and $\boldsymbol{p}_i^H \in \mathcal{H}_{\kappa_H}^{d_H}$, $z_i^H \in \mathcal{T}_{\boldsymbol{p}_i^H} \mathcal{H}_{\kappa_H}^{d_H}$, $\boldsymbol{p}_i^S \in \mathcal{S}_{\kappa_S}^{d_S}$, $z_i^S \in \mathcal{T}_{\boldsymbol{p}_i^S} \mathcal{H}_{\kappa_H}^{d_H}$. Accordingly, Riemannian metric of the product bundle is yielded as $\mathfrak{g}_x^{\mathcal{P}} = \mathfrak{g}_x^{\kappa_H} \oplus \mathbf{I}_{d_H+1} \oplus \mathfrak{g}_x^{\kappa_S} \oplus \mathbf{I}_{d_H+1}$, where $\mathbf{I}_{d_H+1}$ is the $(d_H + 1)$-dimensional identity matrix, and $\oplus$ denotes the direct sum among matrices. In Eq. (2), hyperbolic bundle $\mathcal{H}_{\kappa_H}^{d_H} \otimes \mathcal{T}\mathcal{H}_{\kappa_H}^{d_H}$ and hyperspherical bundle $\mathcal{S}_{\kappa_S}^{d_S} \otimes \mathcal{T}\mathcal{S}_{\kappa_S}^{d_S}$ are responsible for trees and cycles, respectively. Our framework is applicable to multiple bundles with any curvatures, and we use the two-bundle product for simplicity. In this paper, we opt for the unified formalism $\mathcal{L}_\kappa^d$ in Eq. (1) for hyperbolic and hyperspherical spaces.

### 3.1.2 *Deriving Riemannian Operations.*

Before designing the neural architecture, we derive a closed-form Riemannian linear operation and introduce a geometric midpoint for mathematical preparation. (Proofs are given in Appendix B.) In Riemannian geometry, the operation output is required to remain on the manifold, i.e., manifold preserving. Previous works typically meet this requirement by involving an additional tangent space with the exponential/logarithmic maps [3, 23]. However, the lack of isometry and possible mapping error [44] motivate a fully Riemannian formulation. We formulate the linear operation with matrix-left-multiplication. The operation parameterized by $\boldsymbol{W}$ is derived as

$$\forall \boldsymbol{x} = \begin{bmatrix} x_t \\ \boldsymbol{x}_s \end{bmatrix} \in \mathcal{L}_\kappa^d, \quad f_{\boldsymbol{W}}(\boldsymbol{x}) = \begin{bmatrix} 1 & \mathbf{0}^\top \\ \mathbf{0} & \alpha\boldsymbol{W} \end{bmatrix} \begin{bmatrix} x_t \\ \boldsymbol{x}_s \end{bmatrix}, \quad (3)$$

where re-scaling factor is defined as $\alpha = \frac{\sqrt{\kappa^{-1} - sgn(\kappa)x_t^2}}{\|\boldsymbol{W}\boldsymbol{x}_s\|^2}$ and $\|\cdot\|$ denotes the $L2$ norm. The theoretical guarantee is given below.

THEOREM 1 (MANIFOLD-PRESERVING OF PROPOSED OPERATION). *Given $\boldsymbol{x} \in \mathcal{L}_\kappa^{d_1}$ and $\kappa \neq 0$, $f_{\boldsymbol{W}}(\boldsymbol{x}) \in \mathcal{L}_\kappa^{d_1}$ preserves on the manifold with any $\boldsymbol{W} \in \mathbb{R}^{d_1 \times d_1}$, and $f_{\boldsymbol{W}}(\boldsymbol{x}) \in \mathcal{L}_\kappa^{d_2}$ holds for any $\boldsymbol{W} \in \mathbb{R}^{d_1 \times d_2}$.*

We notice that Chen et al. [6] and Yang et al. [42] propose linear operation in fully Riemannian fashion recently, but none of them allows for operating in any constant curvature. The aggregation is typically given as an arithmetic mean in Euclidean spaces [13, 46]. With a set of points and their weights $\{\boldsymbol{x}_i, \nu_i\}_{i \in \Omega}$, $\boldsymbol{x}_i \in \mathcal{L}_\kappa^d$, $\nu_i \in \mathbb{R}$, the arithmetic mean in CCS takes the form of

$$mid_\kappa(\{\boldsymbol{x}_i, \nu_i\}_{i \in \Omega}) = \frac{1}{\sqrt{|\kappa|}} \sum_{i \in \Omega} \frac{\nu_i \boldsymbol{x}_i}{\left| \|\sum_{j \in \Omega} \nu_j \boldsymbol{x}_j\|_\kappa \right|}, \quad \kappa \neq 0, \quad (4)$$

with the definition of $\|\boldsymbol{x}\|_\kappa^2 = \langle \boldsymbol{x}, \boldsymbol{x} \rangle_\kappa$. We demonstrate the fact that the mean in Eq. (4) is the geometric midpoint on the manifold.

THEOREM 2 (ARITHMETIC MEAN AS GEOMETRIC MIDPOINT). *The arithmetic mean in Eq. (4) is on the manifold $\boldsymbol{c} = mid_\kappa(\{\boldsymbol{x}_i, \nu_i\}_{i \in \Omega}) \in \mathcal{L}_\kappa^d$, and is the geometric midpoint $\boldsymbol{c} = \arg\min_{\boldsymbol{c} \in \mathcal{L}_\kappa^d} \sum_{i \in \Omega} \nu_i d_\kappa^2(\boldsymbol{c}, \boldsymbol{x}_i)$ w.r.t. the squared distance $d$.*

### 3.1.3 *Vocabulary Learning Module.*

This module focuses on the substructure, with the objective of embedding the structural vocabulary into the constant curvature spaces. In other words, we are interested in how to place a tree (or cycle) in the hyperbolic (or hyperspherical) space. To this end, we propose a **Cross-geometry Attention** to learn node coordinates in the substructure. We elaborate the formulation with the tree in hyperbolic factor. In hyperbolic manifold, we propose to update a tree in bottom-up fashion, and thus this problem is reduced to induce the node coordinate from its descendant nodes. The node coordinate is given by the attentional aggregation with attentional weights as follows,

$$\boldsymbol{v}_i = mid_\kappa(\{\boldsymbol{v}_j, \alpha_{ij}\}_{(i,j) \in \Omega}) \in \mathcal{L}_\kappa^d, \quad \boldsymbol{v}_j \in \mathcal{L}_\kappa^d, \quad (5)$$

$$\alpha_{ij} = \frac{\exp(\phi([\boldsymbol{q}_i || \boldsymbol{k}_j]))}{\sum_{(i,t) \in \Omega} \exp(\phi([\boldsymbol{q}_i || \boldsymbol{k}_t]))}, \quad (6)$$

where $j$ is the descendant node of $i$, and we slightly abuse $j$ to include the coordinate information of $i$ itself. In cross-geometry attention, the key, query and value are derived with the Riemannian linear operation $\boldsymbol{k}_i = f_V(\boldsymbol{p}_i^H)$, $\boldsymbol{q}_i = f_Q(\boldsymbol{p}_i^S)$ and $\boldsymbol{v}_i = f_V(\boldsymbol{p}_i^H)$, respectively. $\phi$ can be any function that returns a scalar. As a result, the node coordinate $\boldsymbol{p}_i^H$ is updated as $\boldsymbol{v}_i$. Note that, the query value is given from $\mathcal{S}_\kappa^d$ so as to leverage the compensatory information of the other geometry. (Compared to performing attention in a single geometry, the superiority of our design is evaluated in the Ablation Study.) In addition, the proposed aggregation is **unidirectional** which is different from that of traditional bidirectional aggregations in graph model [13, 20, 46]. In traditional aggregations, each node considers its information in the neighborhood, and vice versa. However, as in Eq. (5), each node receives the coordinates of the descendant nodes to locate itself on the manifold, while the reverse information path does not exist, that is, the node's coordinate is not affected by the ancestor node in bottom-up construction.

Similarly, the cycle is refined on hyperspherical manifold where the node coordinate is updated by the two nodes connecting it. The unidirectional path is from neighboring nodes to the center.

**Comparison to Graph Transformers.** Despite the differences in generalization capacity, the proposed architecture is fundamentally different from that of graph transformers, which conducts the bidirectional attention to all nodes, typically in Euclidean space. On the contrary, the proposed attention is unidirectional and is performed over graph substructure in account of its Riemannian geometry. We notice that Yang et al. [42] introduces a transformer net (Hypformer) very recently. However, we consider each substructure in the corresponding Riemannian manifold with cross-geometry keys, while Hypformer places the input as a whole in hyperbolic space.

### 3.1.4 *Global Learning Module.*

Sampling multiple substructures from the graph, this module examines the entire graph to learn node encodings from a global perspective. This objective is achieved by the following two phases.

Firstly, we study the node encoding at substructure level. Note that, node encodings live in the tangent bundle surrounding the manifold, where the tangent space of one point is incompatible with that of another point [29]. Hence, existing message passing formulations (e.g., GCN [20], Constant Curvature GCN [1]) cannot be used due to space incompatibility. To bridge this gap, we propose a **Bundle Convolution** for message passing over tangent bundles. The unified formalism for arbitrary curvature is derived as,

$$BC_{\boldsymbol{p}_t}(\{\boldsymbol{p}_i, z_i\}_{i \in \Lambda}) = \sum_{i \in \Lambda} \left( \alpha_{it} z_i - \frac{\kappa\alpha_{it} \langle z_i, \boldsymbol{p}_t \rangle_\kappa}{1 + \kappa\langle \boldsymbol{p}_i, \boldsymbol{p}_t \rangle_\kappa} (\boldsymbol{p}_i + \boldsymbol{p}_t) \right), \quad (7)$$

**Figure 2: An illustration of bundle convolution.**

where $\Lambda$ is the node set of the substructure and the attentional weight $\alpha_{it}$ is derived by Eq. (6) over the substructure. The rationale of resolving space incompatibility lies in parallel transport, a canonical way to connect different tangent spaces.

PARALLEL TRANSPORT. *In Riemannian geometry, the parallel transport w.r.t. the Levi-Civita connection $PT_{x \to y}$ transports a vector in $\boldsymbol{v} \in \mathcal{T}_x \mathcal{M}$ to another tangent space $\mathcal{T}_y \mathcal{M}$ with a linear isometry along the geodesic between $\boldsymbol{x}, \boldsymbol{y} \in \mathcal{M}$.*

Accordingly, Eq. (7) can be explained as the following process. The encodings are parallel transported to the tangent space of the target point, in which message passing is subsequently conducted. A visual illustration is given in Fig. 2, where $a$ is the target point whose encoding is to be updated. The advantage of bundle convolution is that we consider the encoding of the global structure while encapsulating the local geometry of the manifold. (The detailed derivation is provided in Appendix C.)

Secondly, we obtain the output node encoding at the graph level. As in **Fig. 1(c)**, there are three cycles (and trees) containing node $a$, and we aim to align the coordinates of node $a$ and obtain the graph-level node encoding. With $K$ samples of $a$, the alignment is given by the geometric midpoint of coordinates on the manifold, $\boldsymbol{p}_a = mid_\kappa(\{\boldsymbol{p}_{a_i}\}_{i=1,\cdots,K}) \in \mathcal{L}_\kappa^d$, where aggregation weight is set as 1. Then, node encodings of each sample are parallel transported to the tangent space of the midpoint. Consequently, the graph-level node encoding is derived as $z_a = BC_{\boldsymbol{p}_a}(\{\boldsymbol{p}_{a_i}, z_{a_i}\}_{i=1,\cdots,K})$.

***On Stacking Multiple Layers.*** The main advantage is to enlarge the receptive field. For example, a node in the tree is updated by its first-order descendant nodes with one layer, as in Eq. (5), and is further affected by the second-order descendant nodes with another layer. By stacking multiple layers, a node can perceive a larger region in the substructure, while simultaneously broadening its global view by calculating the agreement across the entire graph.

***Comparison to Previous Riemannian Graph Models.*** Our idea is fundamentally different from that of previous Riemannian models. All of them explore advanced techniques to embed nodes in the **manifold**, either in a single CCS [1], the product [7, 12], or the quotient [21, 41], to the best of our knowledge. Note that, we seek node encodings in the **tangent bundle**, considering the structural vocabulary and global information.

## 3.2 Geometric Contrastive Learning

A foundation model requires self-supervised learning to acquire shared knowledge that is not tied to specific annotations. Contrastive learning has become an effective method for self-supervised learning, but it is nontrivial for graphs; for example, graph augmentation to generate contrastive views is not as easily accessible

---

**Algorithm 1:** Training Algorithm of `RiemannGFM`

**Input:** pre-training graphs, Hyperparameters of the product bundle and `RiemannGFM`.

**Output:** Model parameters of `RiemannGFM`

1  Initialize node encodings and node coordinates on CCSs;
2  Sample substructures according to the structural vocabulary;
3  **while** *model not converged* **do**
4      **for** *each substructure in each geometry* **do**
5          Conduct the cross-geometry attention in Eq. (5) to update node coordinates;
6          Conduct the bundle convolution in Eq. (7) to update node encodings in the substructure;
7      Induce the node encodings with global view with the geometric midpoint for each geometry;
8      Generate the hyperbolic and hyperspherical views;
9      Compute the geometric contrastive loss with Eq. (8);
10     Update model parameters via gradient descent.

---

as cropping/rotating of images. Thanks to the diverse structural geometries in our design, they offer different views for graph contrastive learning (i.e., hyperbolic view and hyperspherical view).

Here, we introduce the geometric contrastive objective on the product bundle for the self-supervised learning of our model, free of graph augmentation. Concretely, the node encoding in the tangent space acts as the geometric view of the corresponding manifold. Thus, the remaining ingredient is a score function that contrasts positive and negative samples, and the challenge lies in the incompatibility between different geometries. To bridge this gap, we consider a shared tangent space of the north pole of Lorentz/Spherical model. Thus, the geometric contrast is given as follows,

$$\mathcal{J}(H, S) = -\sum_{i=1}^{N} \log \frac{\exp(\langle PT_{\boldsymbol{p}_i^H \to \boldsymbol{o}}(\boldsymbol{z}_i^H), PT_{\boldsymbol{p}_i^S \to \boldsymbol{o}}(\boldsymbol{z}_i^S)\rangle)}{\sum_{j=1}^{N} \exp(\langle PT_{\boldsymbol{p}_i^H \to \boldsymbol{o}}(\boldsymbol{z}_i^H), PT_{\boldsymbol{p}_j^S \to \boldsymbol{o}}(\boldsymbol{z}_j^S)\rangle)}. \quad (8)$$

The overall objective is formulated as $\mathcal{J}_0 = \mathcal{J}(H, S) + \mathcal{J}(S, H)$, where $N$ is the number of nodes. Though the geometric contrast is done over node encodings in tangent spaces, the parameters of factor manifolds are encapsulated in the parallel transport among tangent spaces. The training procedure is summarized in Algorithm 1, whose **computational complexity** is yielded as $O(|\mathcal{V}|^2 + |\mathcal{E}|)$, where $\mathcal{V}$ and $\mathcal{E}$ are the node set and edge set, respectively, and the proposed model supports minibatch training. Finally, `RiemannGFM` **is capable of generating informative node encodings for an arbitrary graph, with the shared structural knowledge of the graph domain learned in Riemannian geometry.**

## 4 Experiment

In this section, we aim to answer the following research questions:

- **RQ1.** How does `RiemannGFM` perform in cross-domain transfer learning?
- **RQ2.** How significant is embedding structural vocabulary into Riemannian geometry, rather than Euclidean ones?
- **RQ3.** How effective is `RiemannGFM` under few-shot learning?
- **RQ4.** How expressive is the structural knowledge learned by `RiemannGFM`?
- **RQ5.** How does the pre-training dataset impact `RiemannGFM`?

**Table 1: Cross-domain transfer learning performance on Citeseer, Pubmed, GitHub and Airport datasets. Node classification and link prediction results are reported. The best results are in boldfaced.**

| Method | Node Classification Results | | | | | | | | Link Prediction Results | | | | | | | |
| | Citeseer | | Pubmed | | GitHub | | Airport | | Citeseer | | Pubmed | | GitHub | | Airport | |
| | ACC | F1 | ACC | F1 | ACC | F1 | ACC | F1 | AUC | AP | AUC | AP | AUC | AP | AUC | AP |
|---|---|---|---|---|---|---|---|---|---|---|---|---|---|---|---|---|
| GCN [3] | 70.30 | 68.56 | 78.90 | 77.83 | 85.68 | 84.34 | 50.80 | 48.09 | 90.70 | 92.91 | 91.16 | 89.96 | 87.48 | 85.34 | 92.37 | 94.24 |
| SAGE [13] | 68.24 | 67.60 | 77.57 | 73.61 | 85.12 | 77.36 | 49.16 | 47.57 | 87.29 | 89.03 | 87.02 | 86.85 | 79.13 | 81.21 | 92.17 | 93.56 |
| DGI [37] | 71.30 | 71.02 | 76.60 | 76.52 | 85.19 | 84.10 | 50.10 | 49.56 | 96.90 | 97.05 | 88.39 | 87.37 | 86.39 | 86.61 | 92.50 | 91.63 |
| GraphMAE2 [14] | **73.40** | 71.68 | **81.10** | 79.78 | 85.23 | 83.34 | 52.34 | 49.02 | 92.75 | 89.23 | 89.46 | 85.37 | 87.11 | 86.23 | 88.23 | 90.23 |
| GCOPE [47] | 65.33 | 62.34 | 74.15 | 74.33 | 82.29 | 72.89 | 39.96 | 36.40 | 88.60 | 83.03 | 90.84 | 86.45 | 82.16 | 83.22 | 86.17 | 84.91 |
| OFA [22] | 58.32 | 65.41 | 74.40 | 72.42 | - | - | - | - | 82.62 | 83.74 | 92.26 | 91.36 | - | - | - | - |
| GraphAny [48] | 66.10 | 63.01 | 76.10 | 70.12 | 79.45 | 77.19 | 47.98 | 46.88 | - | - | - | - | - | - | - | - |
| OpenGraph[40] | 58.58 | **76.78** | 58.40 | 56.49 | 30.16 | 30.16 | 40.45 | 38.28 | 76.78 | 77.35 | 70.02 | 72.23 | 86.72 | 87.42 | 85.32 | 83.25 |
| LLaGA [5] | 59.00 | 56.91 | 71.21 | 63.38 | 53.33 | 54.17 | 38.49 | 39.89 | 86.26 | 83.35 | 84.04 | 76.48 | 71.25 | 70.63 | 77.90 | 74.30 |
| RiemannGFM | 66.38 | 66.41 | 76.20 | 75.83 | **85.96** | **85.57** | **55.29** | **53.27** | **99.40** | **98.42** | **94.12** | **91.64** | **89.18** | **93.52** | **93.68** | **96.07** |

(GFM group: GCOPE, OFA, GraphAny, OpenGraph, LLaGA)

**Table 2: Geometric ablation on Citeseer, Pubmed, and Airport datasets. Link prediction results are reported in terms of AUC (%). The results are given in the form of mean±std. $\mathcal{R}_0^{32}$ denotes the Euclidean space.**

| Trees | Cycles | Citeseer | Pubmed | Airport |
|---|---|---|---|---|
| $\mathcal{H}_{-1}^{32}$ | $\mathcal{S}_1^{32}$ | **99.40 ± 0.06** | **94.12 ± 1.38** | **93.68 ± 0.09** |
| $\mathcal{H}_{-1}^{32}$ | $\mathcal{R}_0^{32}$ | 98.48 ± 0.32 | 92.41 ± 2.14 | 92.22 ± 1.43 |
| $\mathcal{H}_{-1}^{32}$ | $\mathcal{H}_{-1}^{32}$ | 98.21 ± 0.45 | 92.32 ± 2.57 | 91.79 ± 1.58 |
| $\mathcal{H}_{-1}^{32}$ | $\mathcal{S}_1^{32}$ | **99.40 ± 0.06** | **94.12 ± 1.38** | **93.68 ± 0.09** |
| $\mathcal{R}_0^{32}$ | $\mathcal{S}_1^{32}$ | 98.72 ± 0.08 | 92.43 ± 1.53 | 92.51 ± 0.18 |
| $\mathcal{S}_1^{32}$ | $\mathcal{S}_1^{32}$ | 98.85 ± 0.09 | 92.88 ± 1.47 | 92.85 ± 0.23 |

## 4.1 Experimental Setups

*4.1.1 **Datasets**.* The experiments are conducted on a diversity of datasets. Specifically, we include two text-attributed graphs (the popular Citeseer and Pubmed [43]), one mix-attributed graph (GitHub [31], whose attributes consist of the location, starred repositories, employer, and e-mail address), and one non-attributed graph (Airports [30], containing airports and connections among them).

*4.1.2 **Baselines**.* We include 9 strong baselines categorized into three groups: The first group is the **vanilla GNNs**: GCN [20] and GraphSAGE [13] with the end-to-end training paradigm. The second group consists of **self-supervised graph learning models**: DGI [37] and GraphMAE2 [14]. The third group is **graph foundation models**, including OFA [22], GCOPE [47], GraphAny [48], LLaGA [5] and OpenGraph [40]. The proposed model is named RiemannGFM, considering the shared structural knowledge. (Dataset and baseline description is detailed in Appendix E.)

*4.1.3 **Evaluation Metrics**.* We evaluated the comparison methods by both node classification and link prediction tasks. For node classification, we employ two popular metrics of classification accuracy (ACC) and weighted F1-score (F1), while for the link prediction, the mean Area Under the Receiver Operating Characteristic curve (AUC-ROC) and average precision (AP) are utilized. To ensure statistical robustness and fair comparison, we conducted 10 independent runs for each model and reported the mean performance along with the standard deviation. All experiments are conducted on a server equipped with an NVIDIA GeForce RTX 4090D GPU (24GB VRAM) utilizing CUDA 11.8, an AMD EPYC 9754 128-Core Processor (18 vCPUs allocated), 60GB of RAM.

*4.1.4 **Model Configuration and Reproducibility**.* First, we introduce the model initialization of the proposed RiemannGFM as follows. The input node encoding is given from the Laplacian matrix, encapsulating the structural information. Note that, we leverage the eigenvectors of $K$ largest eigenvalues, which normalizes different graph datasets with a predefined $K$. Correspondingly, node coordinates are initialized on the constant curvature spaces by the exponential map with the reference point of the north pole. Second, on model configuration, we utilize the standard curvature for hyperbolic and hyperspherical spaces, and the dimension is set as 32 by default. That is, RiemannGFM is instantiated on the product bundle of $\left(\mathcal{H}_{-1}^{32} \otimes \mathcal{T}\mathcal{H}_{-1}^{32}\right) \otimes \left(\mathcal{S}_1^{32} \otimes \mathcal{T}\mathcal{S}_1^{32}\right)$. The RiemannGFM consists of two universal Riemannian layers. As for the structural vocabulary, trees are in hyperbolic space, while hyperspherical space accommodates cycles of node triangles and quadruples. The parameters are optimized by Adam [19] with the learning rate and dropout rate are tuned with grid search. **Codes** are provided in the anonymous link of https://anonymous.4open.science/r/Geo-GFM-1603. (Please refer to Appendix E for further implementation notes.)

## 4.2 Results and Discussion

*4.2.1 **Cross-domain Transfer Learning Performance (RQ1)**.* The results for both node classification and link prediction are summarized in Table 1. The proposed RiemannGFM is pre-trained on the datasets of ogbn-arxiv [15], Physics [33], Amazon-Computers [33] and its classification head is the same as that of popular GFMs [39, 40, 47]. Note that, OFA [22] cannot work on the graphs without textual attributes (i.e., Github and Airport datasets). GraphAny [48] generates classification logistics only, and thereby cannot be utilized for link prediction. As shown in Table 1, in the link prediction task, the proposed RiemannGFM consistently achieves the best results among GFMs and specialized models, i.e., GCN, SAGE, DGI and GraphMAE2. The structural vocabulary embedded in Riemannian manifolds contributes to our success, which is further discussed in RQ3. In node classification tasks, the proposed RiemannGFM achieves the best results on the graphs without textual attributes. On text-attributed graphs, RiemannGFM still obtain comparable performance to previous GFMs. This shows the importance of building GFM that can capture the structural information.

*4.2.2 **The Connection between Structural Vocabulary and Constant Curvature Spaces (CCSs) (RQ2)**.* Given the significance of structural vocabulary, we are interested in which CCS is

**Table 3: Few-shot learning performance on Citeseer, Pubmed, GitHub, and Airport datasets. The best results are in boldfaced.**

| Setting | Method | Node Classification Results | | | | | | | |
|---|---|---|---|---|---|---|---|---|---|
| | | Citeseer | | Pubmed | | GitHub | | Airport | |
| | | ACC | F1 | ACC | F1 | ACC | F1 | ACC | F1 |
| 1-Shot | DGI [37] | 37.40 ±9.98 | 32.29 ±12.17 | 39.29 ±3.79 | 34.76 ±5.12 | 59.90 ±4.89 | 55.48 ±9.73 | 30.63 ±6.14 | 17.57 ±7.28 |
| | GraphMAE2 [14] | 34.62 ±4.23 | 31.34 ±1.21 | 39.10 ±6.45 | 35.97 ±8.83 | 52.47 ±3.98 | 50.25 ±4.78 | 29.89 ±5.45 | 20.27 ±6.51 |
| | OFA [22] | 37.58 ±10.51 | 30.90 ±2.85 | 39.80 ±0.74 | 27.54 ±3.05 | - | - | - | - |
| | GCOPE [47] | 36.03 ±4.63 | 31.89 ±4.54 | 37.36 ±4.21 | 23.64 ±3.80 | 56.07 ±5.09 | 43.89 ±6.22 | 26.09 ±0.99 | 18.05 ±4.95 |
| | OpenGraph [48] | 20.60 ±2.43 | 18.30 ±1.01 | 43.58 ±1.12 | 35.39 ±1.03 | 22.19 ±0.03 | 40.32 ±0.65 | 31.94 ±2.99 | 23.38 ±2.13 |
| | LLaGA [5] | 18.10 ±2.03 | 14.57 ±0.97 | 35.68 ±1.58 | 33.48 ±1.34 | 26.67 ±1.96 | 28.89 ±2.54 | 23.53 ±2.02 | 19.17 ±2.31 |
| | **RiemannGFM** (Ours) | **38.02** ±9.45 | **32.42** ±9.87 | **45.24** ±3.55 | **37.87** ±6.56 | **77.83** ±4.53 | **72.46** ±7.54 | **32.61** ±4.74 | **27.18** ±7.46 |
| 5-Shot | DGI [37] | 46.48 ±1.32 | 43.62 ±1.49 | 51.38 ±4.05 | 50.90 ±3.86 | 65.38 ±0.13 | 64.55 ±0.28 | 37.61±6.41 | 28.85 ±6.16 |
| | GraphMAE2 [14] | 47.12 ±4.01 | 44.71 ±1.88 | 53.04 ±4.11 | 47.74 ±4.37 | 62.22 ±2.19 | 60.88 ±7.42 | 37.09 ±6.02 | 29.11 ±2.02 |
| | OFA [22] | 31.90 ±4.27 | 23.04 ±0.83 | 36.72 ±9.40 | 24.43 ±6.12 | - | - | - | - |
| | GCOPE [47] | 43.48 ±9.55 | 38.65 ±9.07 | 46.35 ±9.59 | 44.85 ±9.36 | 73.26 ±2.07 | 63.13 ±1.57 | 33.18 ±2.38 | 27.71 ±6.09 |
| | OpenGraph [48] | 29.30 ±1.81 | 27.46 ±1.41 | 37.52 ±2.52 | 35.51 ±3.24 | 24.32 ±0.45 | 42.30 ±0.04 | 33.51 ±3.55 | 23.74 ±2.15 |
| | LLaGA [5] | 21.60 ±2.11 | 24.89 ±2.32 | 32.02 ±1.85 | 35.84 ±1.96 | 58.33 ±1.35 | 56.86 ±1.26 | 32.86 ±1.24 | 30.50 ±1.23 |
| | **RiemannGFM** (Ours) | **53.46** ±4.17 | **51.89** ±4.60 | **66.18** ±5.99 | **64.56** ±9.38 | **84.19** ±1.05 | **83.13** ±1.89 | **38.72** ±5.98 | **33.40** ±5.66 |

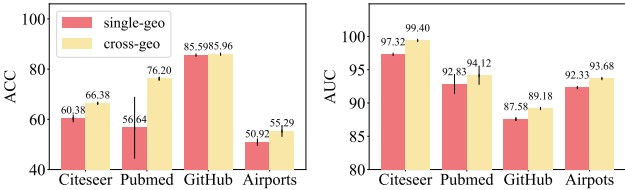

(a) Node Classification  (b) Link Prediction

**Figure 3: Ablation on cross-geometric attention.**

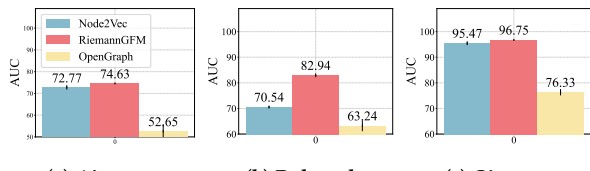

(a) Airports  (b) Pubmed  (c) Citeseer

**Figure 4: Link prediction results with structural knowledge**

suitable to model trees and cycles. Theoretically, hyperbolic space aligns with the tree as evidenced in the consistency of volume growth [27], while hyperspherical space acts as the geometric analogy of cycles according to the common rotational invariant [29]. Here, we empirically investigate our choice by geometric ablation. To be specific, we place trees and cycles in hyperbolic, hyperspherical, and Euclidean spaces, respectively, and summarize and link prediction results in Table 2. It achieves the best performance when trees are embedded in hyperbolic space, and cycles in hyperspherical space, aligning with our choices.

*4.2.3 Ablation Study on Cross-geometry Attention*. We conduct an ablation study to evaluate the effectiveness of cross-geometry attention, whose query vector is in the counterpart CCS of key and value vector. To this end, we introduce a model of a single-geometry variant, which utilizes the key, query, and value vectors in the same CCS. Fig. 3 collects node classification and link prediction results on different datasets. The cross-geometry attention consistently outperforms the single-geometry variant, demonstrating the effectiveness of our design.

*4.2.4 Few-shot Learning Performance (RQ3)*. We report the node classification results under 1-shot and 5-shot learning in Table 3. The self-supervised models (i.e., DGI and GraphMAE2) are trained merely on the few-shot set, while the GFMs undergo model pre-training and are subsequently fine-tuned on the few-shot, following the setting of [40]. (Further details are introduced in Appendix E.) As shown in Table 3, we observe an interesting phenomenon: OpenGraph and LLaGA exhibit negative transfer on GitHub and

Airport datasets. They leverage the LLM and enjoy shared knowledge among textural attributes. However, it becomes problematic when transferring such knowledge to mixed attributes (e.g., the numbers and addresses in GitHub) or to the graphs without attributes. This highlights the limitation of coupling graph transfer with textual attributes, and thus supports our motivation to explore common structures for better universality.

*4.2.5 On the Expressiveness of Structural Knowledge (RQ4)*. We show that, despite the universality of structures in the graph domain, the structural knowledge itself presents promising expressiveness. Concretely, we examine the link prediction performance of RiemannGFM, compared to node2vec [11] and GFMs, i.e., OpenGraph [40]. In this case, node encodings generated from pre-trained RiemannGFM are utilized for link prediction; that is, we leverage the structural knowledge learned on pre-training datasets and do not include attributes of the target graph, while node2vec is trained on the target datasets. The results are given in Fig. 4. Note that, structural knowledge of RiemannGFM acquires competitive even superior results to specialized model for specific graphs and GFMs incorporated with attributes, showing its promising expressiveness.

*4.2.6 Impact of Pre-training Datasets (RQ5)*. To further investigate the transferability, we study the performance of RiemannGFM with different pre-training datasets. We adopt Flicker [45], Amazon-Computers [33], and WikiCS [26] as pre-training datasets respectively, and report the results in Table 4. We find that: RiemannGFM shows more stable performance over different pre-training graphs, that is, the pre-training datasets have limited impact on our model. However, GCOPE and OpenGraph have higher requirements for

**Table 4: Cross-domain link prediction prformance on different pre-training datasets.**

| Pre-training | Method | Testing Datasets | | |
|---|---|---|---|---|
| | | Citeseer | Pubmed | Airport |
| Flickr | OpenGraph | $65.16_{\pm 2.54}$ | $50.66_{\pm 2.14}$ | $86.42_{\pm 2.15}$ |
| | GCOPE | $84.20_{\pm 0.12}$ | $85.60_{\pm 0.62}$ | $84.54_{\pm 1.09}$ |
| | RiemannGFM | $\mathbf{99.33_{\pm 1.36}}$ | $\mathbf{92.51_{\pm 0.73}}$ | $\mathbf{93.52_{\pm 0.06}}$ |
| AComp | OpenGraph | $60.16_{\pm 3.21}$ | $60.56_{\pm 2.24}$ | $87.31_{\pm 0.56}$ |
| | GCOPE | $85.01_{\pm 1.00}$ | $84.63_{\pm 1.30}$ | $85.21_{\pm 0.89}$ |
| | RiemannGFM | $\mathbf{99.40_{\pm 1.72}}$ | $\mathbf{92.75_{\pm 0.61}}$ | $\mathbf{93.21_{\pm 0.03}}$ |
| WikiCS | OpenGraph | $89.64_{\pm 3.45}$ | $72.24_{\pm 4.24}$ | $86.89_{\pm 3.12}$ |
| | GCOPE | $88.51_{\pm 0.47}$ | $89.07_{\pm 0.58}$ | $86.09_{\pm 1.14}$ |
| | RiemannGFM | $\mathbf{99.31_{\pm 0.02}}$ | $\mathbf{92.47_{\pm 0.73}}$ | $\mathbf{93.17_{\pm 0.07}}$ |

pre-training datasets. Pre-training on similar domains enhances the performance of downstream tasks. For example, when tested on the citation network of Citeseer, OpenGraph achieves 89.64% with the pre-training dataset WikiCS (citation network), but has performance loss with pre-training datasets of other domains (65.16% on Flickr and 60.16% on AComp). GCOPE is potentially affected by differences in attribute distribution across different domains. The reason is two-fold: 1) The structural transferability of RiemannGFM enjoys greater universality, especially when attributes show obvious disparities across in different domains. 2) The structural vocabulary is proposed to learn the shared structural knowledge underlying the graph domain and is not tied to any specific structures.

*4.2.7 Visualization and Discussion.* Here, we visualize the node encoding of Cora via t-sne in Fig. 5, where different colors denote different node classes. Fig. 5(b) shows the results of GCN, while the visualization of pre-trained RiemannGFM in Fig. 5(c) is given by its node encodings in the shared tangent space of the north pole of Lorentz/Spherical model. It shows that the encodings of pre-trained RiemannGFM are more separable than those of a specialized graph model, demonstrating the expressiveness of the knowledge learned in RiemannGFM. (**Additional results are given in Appendix F.**)

## 5 Related Work

***Graph Neural Network & Self-supervised Graph Learning.*** Popular GNNs include graph convolutional nets and graph transformers. The former conducts neighborhood aggregation with layer-wise message passing [8, 13, 38], while the latter leverages a transformer-like encoder [18]. Both of them are typically paired with a classification head or reconstructive loss on a specified graph. Thus, a major shortcoming of traditional graph models is their limited generalization capability. Self-supervised learning has been integrated into GNNs in recent years [14, 37]. Instead of coupling GNNs with downstream tasks, self-supervised learning conducts parameter training from the graph data itself via specialized pretext tasks. However, graph augmentation for self-supervised learning is nontrivial [17, 50], and the parameters trained on one graph cannot be directly applied to another owing to the difference in attribute distribution. In other words, existing graph models lack the universality, and are still far from being a foundation model.

***Graph Foundation Model.*** Recent efforts are generally categorized into two groups. The first group enhances the vanilla GNNs to achieve better generalization capacity, e.g., unifying the downstream tasks with graph prompt [35], and training on multi-domain

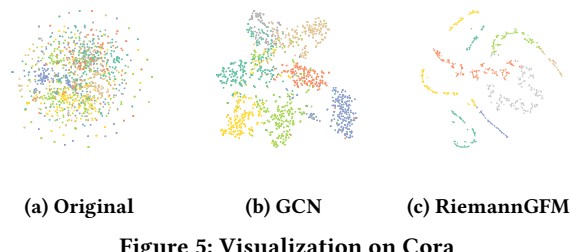

| (a) Original | (b) GCN | (c) RiemannGFM |
|---|---|---|

**Figure 5: Visualization on Cora**

graphs with coordinators [47]. Zhao et al. [48] generalize SGC [38] for node classification on any graph. The second group adapts LLM for analyzing graphs. LLaGA [5] tailors graphs for the language model with node sequences, generated via graph translation, while OFA [22] unifies different graph data by the language description of nodes and edges. OpenGraph [40] re-frames textual attributes into language with a hierarchy. Very recently, Xia and Huang [39] propose a mixture of graph experts. Existing models typically struggle to maintain the performance on graphs without textual attributes. Also, they model graphs in Euclidean space, and tend to trivialize the structural complexity. Distinguishing from the prior studies, we consider graphs in Riemannian geometry, and design the first GFM exploring graph substructures (structural vocabulary), to the best of our knowledge.

***Riemannian Manifold & Graphs.*** Euclidean space has been the workhorse of graph representation learning for decades, and Riemannian manifolds are emerging as exciting alternatives in recent years. Among Riemannian manifolds, hyperbolic space is well recognized for its alignment with tree-like (hierarchical) structures, and hyperbolic GNNs show superior results to Euclidean counterparts [3, 10]. The geometric analogy of cycles is hyperspherical space, whose advantage of embedding cyclical structures is reported [25, 49]. Bachmann et al. [1] formulate a graph convolutional net in constant curvature spaces. We notice that the product manifold has been introduced to study graphs recently, and advanced techniques are proposed for node embedding [7, 12]. However, all of them lack the generalization capability to unseen graph structures, and consider node embeddings on the manifold, while we introduce the notion of tangent bundle to model graphs for the first time. So far, the potential of Riemannian geometry has not yet been released on GFM, and we are dedicated to bridging this gap.

## 6 Conclusion

This work opens a new opportunity to build GFM with a shared structural vocabulary of the graph domain. Our main contribution is the discovery of tree-cycle vocabulary with the inherent connection to Riemannian geometry, and we present a universal pre-training model RemannGFM accordingly. Concretely, we first propose a novel product bundle to incorporate diverse geometries of the vocabulary. On this constructed space, we stack the Riemannian layers where the structural vocabulary, regardless of specific graphs, is learned on Riemannian manifold. This offers the shared structural knowledge for cross-domain transferability, and informative node encodings are generated with the bundle convolution for arbitrary graphs. Extensive experiments show the superior performance of RemannGFM in cross-domain transfer learning and few-shot learning.

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

## A Notation Table

**Table 5: Importation Notations.**

| Notation | Description |
|---|---|
| $\mathcal{M}, \mathfrak{g}$ | A smooth manifold and Riemannian metric. |
| $\mathcal{T}_x \mathcal{M}$ | The tangent space at $x$. |
| $\mathcal{T}\mathcal{M}$ | The tangent bundle surrounding the manifold. |
| $d, \kappa$ | Dimension and curvature. |
| $\mathcal{H}, \mathcal{S}$ | Hyperbolic/Hyperspherical space. |
| $\mathcal{L}$ | A unified formalism of Lorentz/Spherical model. |
| $o$ | North pole of the model space. |
| $\mathcal{G} = (\mathcal{V}, \mathcal{E})$ | A graph with nodes set $\mathcal{V}$ and edges set $\mathcal{E}$. |
| $p \in \mathcal{L}$ | Node coordinate on the manifold. |
| $z \in \mathcal{T}_p \mathcal{L}$ | Node encoding in the tangent space. |
| $\phi : \mathcal{L} \times \mathcal{L} \to \mathbb{R}$ | A parameterized scalar map. |
| $f(\cdot) : \mathcal{L}^m \to \mathcal{L}^n$ | Manifold-reserving linear operation. |
| $[\cdot \|\cdot]$ | Vector concatenation. |
| $\text{Exp}_x(\cdot)$ | The exponential map at point $z$. |
| $\text{Log}_x(\cdot)$ | The logarithmic map at point $z$. |
| $\text{PT}_{x \to y}(\cdot)$ | The parallel transport from $x$ to $y$ |

## B Proofs and Derivations

In this section, we detail the proofs of Theorem 1 and 2, and show the derivation of the proposed bundle convolution.

### B.1 The Proposed Linear Operation

THEOREM 1 (MANIFOLD-PRESERVING OF PROPOSED OPERATION). *Given $x \in \mathcal{L}_\kappa^{d_1}$ and $\kappa \neq 0$, $f_W(x) \in \mathcal{L}_\kappa^{d_1}$ preserves on the manifold with any $W \in \mathbb{R}^{d_1 \times d_1}$, and $f_W(x) \in \mathcal{L}_\kappa^{d_2}$ holds for any $W \in \mathbb{R}^{d_1 \times d_2}$.*

PROOF. We give all the key equations, and do not list all the algebra for clarity. The theorem holds if, with a given curvature $\kappa$, $\kappa \neq 0$, the proposed linear operation satisfies $f_W : \mathcal{L}_\kappa^{d_1} \to \mathcal{L}_\kappa^{d_2}$ for any $W \in \mathbb{R}^{d_1 \times d_2}$. For $x \in \mathcal{L}_\kappa^{d_1}$, we conduct the linear operation,

$$f_W(x) = \begin{bmatrix} 1 & \mathbf{0}^\top \\ \mathbf{0} & \alpha W \end{bmatrix} \begin{bmatrix} x_t \\ x_s \end{bmatrix} = \begin{bmatrix} x_t \\ \alpha W x_s \end{bmatrix}. \qquad (9)$$

With the re-scaling factor $\alpha$ defined as $\frac{\sqrt{\kappa^{-1} - sgn(\kappa)x_t^2}}{\|W x_s\|^2}$, the result is yielded as follows

$$f_W(x) = \begin{bmatrix} x_t \\ \frac{\sqrt{\kappa^{-1} - sgn(\kappa)x_t^2}}{\|W x_s\|^2} W x_s \end{bmatrix}. \qquad (10)$$

Given the equality of $sgn(\kappa)x_t^2 + x_s^\top x_s = \frac{1}{\kappa}$, it is easy to verify the following equality

$$sgn(\kappa)x_t^2 + x_s'^\top x_s' = \frac{1}{\kappa}, \quad x_s' = \frac{\sqrt{\kappa^{-1} - sgn(\kappa)x_t^2}}{\|W x_s\|^2} W x_s, \quad (11)$$

holds for any $W \in \mathbb{R}^{d_1 \times d_2}$. That is, $f_W(x) \in \mathcal{L}_\kappa^{d_2}$ is ensured, completing the proof. □

### B.2 Geometric Midpoint

THEOREM 2 (ARITHMETIC MEAN AS GEOMETRIC MIDPOINT). *The arithmetic mean defined as*

$$mid_\kappa(\{x_i, v_i\}_{i \in \Omega}) = \frac{1}{\sqrt{|\kappa|}} \sum_{i \in \Omega} \frac{v_i x_i}{\|\| \sum_{j \in \Omega} v_j x_j \|_\kappa}, \quad \kappa \neq 0, \quad (12)$$

*is on the manifold $c = mid_\kappa(\{x_i, v_i\}_{i \in \Omega}) \in \mathcal{L}_\kappa^d$, and is the geometric midpoint w.r.t. the squared distance $d$.*

PROOF. The theorem claims two facts. The first is the manifold-preserving of the given arithmetic mean, and the second is the equivalence between the mean and geometric midpoint. We verify the manifold-preserving by manifold definition $sgn(\kappa)c_t^2 + c_s^\top c_s = \frac{1}{\kappa}$, for any $\kappa$, $\kappa \neq 0$.

We elaborate on the geometric midpoint (a.k.a. geometric centroid) before proving the equivalence. Given the set of points of the manifold $x_i \in \mathcal{L}_\kappa^d$ each attached with a weight $v_i$, $i \in \Omega$, the geometric midpoint of squared distance in the manifold $\mathcal{L}_\kappa^d$ is given by the following optimization problem,

$$c = \arg\min_{c \in \mathcal{L}_\kappa^d} \sum_{i \in \Omega} v_i d_\kappa^2(c, x_i), \quad x_i \in \mathcal{L}_\kappa^d. \qquad (13)$$

Now, we derive the geometric midpoint as follows. Recall the fact that $\langle x, x \rangle_\kappa = \frac{1}{\kappa}$ and $d_\kappa^2(x, y) = \frac{2}{\kappa} - 2\langle x, y \rangle_\kappa$. We equivalently transform the minimization of the midpoint in Eq. (13) to the maximization as follows,

$$c = \arg\max_{c \in \mathcal{L}_\kappa^d} \langle \alpha \sum_{i \in \Omega} v_i x_j, c \rangle_\kappa, \qquad (14)$$

where $\alpha$ is a scaling coefficient so that $\alpha \sum_{i \in \Omega} v_i x_j \in \mathcal{L}_\kappa^d$ ($\alpha > 0$). Note that, for any two points $x, y \in \mathcal{L}_\kappa^d$, we have the inequality $\langle x, y \rangle_\kappa < \frac{1}{\kappa}$ and $\langle x, y \rangle_\kappa = \frac{1}{\kappa}$ if and only if $x = y$. That is, we need to find an $\alpha$ to satisfy $\alpha \sum_{i \in \Omega} v_i x_j = c$. Let $\alpha_0 > 0$ satisfies $\alpha_0 \sum_{j \in \hat{N}_i} v_{ij} h_j = c$. As the midpoint is required to live in the manifold, i.e., $\alpha_0 \sum_{i \in \Omega} v_i x_j \in \mathcal{L}_\kappa^d$, we have the following equality

$$\langle \alpha_0 \sum_{i \in \Omega} v_i x_j, \alpha_0 \sum_{i \in \Omega} v_i x_j \rangle_\kappa = \frac{1}{\kappa}, \qquad (15)$$

according to the definition of the manifold in Eq. (1), yielding the scaling coefficient as follows,

$$\alpha_0 = \frac{1}{\sqrt{|\kappa|} \|\| \sum_{i \in \Omega} v_i x_j \|_\kappa} > 0. \qquad (16)$$

Consequently, the geometric midpoint is given as

$$mid_\kappa(\{x_i, v_i\}_{i \in \Omega}) = \frac{1}{\sqrt{|\kappa|}} \sum_{i \in \Omega} \frac{v_i x_i}{\|\| \sum_{j \in \Omega} v_j x_j \|_\kappa}, \qquad (17)$$

completing the proof. □

### B.3 Bundle Convolution

The unified formalism for Bundle Convolution is given as follows,

$$BC_{p_t}(\{p_i, z_i\}_{i \in \Lambda}) = \sum_{i \in \Lambda} \left( \alpha_{it} z_i - \frac{\kappa \alpha_{it} \langle z_i, p_t \rangle_\kappa}{1 + \kappa \langle p_i, p_t \rangle_\kappa} (p_i + p_t) \right). \quad (18)$$

We leverage the equation above to aggregate the node encodings in the corresponding tangent spaces, which span the tangent bundle surrounding the manifold. The key ingredient of the proposed convolution lies in the parallel transport, which solves the incompatibility issue among different tangent spaces.

The parallel transport w.r.t. the Levi-Civita connection $PT_{x \to y}$ transports a vector in $v \in \mathcal{T}_x \mathcal{L}$ to another tangent space $\mathcal{T}_y \mathcal{L}$ with a linear isometry along the geodesic between $x, y \in \mathcal{L}$. Concretely, the unit speed geodesic from $x$ to $v$ is $\gamma_{x,v}(t) = x \cos_\kappa(t) + \frac{1}{\sqrt{|\kappa|}} \sin_\kappa(t) v$, for $t \in [0, 1]$. The generic form in $\mathcal{L}$ is given as

$$PT_{p_i \to p_t}(z_i) = z_i - \frac{\langle Log_{p_i}^\kappa(p_t), z_i \rangle_x}{d_\mathcal{L}(p_i, p_t)} \left( Log_{p_i}^\kappa(p_t) + Log_{p_t}^\kappa(p_i) \right),$$

$$(19)$$

where $\langle a, b \rangle_x = a^\top \mathfrak{g}_x b$ is the inner product at the point $x$, and $\mathfrak{g}_x$

is the Riemannian metric of $\mathcal{L}$ at $\boldsymbol{x}$. Given the logarithmic map with curvature-aware cosine as follows,

$$Log^{\kappa}_{\boldsymbol{p}_i}(\boldsymbol{p}_t) = \frac{\cos_{\kappa}^{-1}(\beta)}{\sqrt{\beta^2 - 1}}(\boldsymbol{p}_t - \beta\boldsymbol{p}_i), \quad \beta = \kappa\langle\boldsymbol{p}_i, \boldsymbol{p}_t\rangle_{\kappa}. \quad (20)$$

The parallel transport in this case is derived as

$$PT_{\boldsymbol{p}_i \to \boldsymbol{p}_t}(\boldsymbol{z}_i) = \boldsymbol{z}_i - \frac{\kappa\langle\boldsymbol{z}_i, \boldsymbol{p}_t\rangle_{\kappa}}{1 + \kappa\langle\boldsymbol{p}_i, \boldsymbol{p}_t\rangle_{\kappa}}(\boldsymbol{p}_i + \boldsymbol{p}_t), \quad \forall\boldsymbol{p}_i, \boldsymbol{p}_t \in \mathcal{L}, \quad (21)$$

where the curvature-aware cosine is defined as $\cos_{\kappa}(\cdot) = \cos(\cdot)$ when $\kappa > 0$, and $\cos_{\kappa}(\cdot) = \cosh(\cdot)$ with $\kappa > 0$, and its superscript $-1$ denotes the inverse function. Therefore, Eq. (18) is given with aggregation over the set $\Lambda$.

## C  Algorithm

We give the pseudocode of cross-geometry attention in Algo. 2.

## D  Riemannian Geometry

A Riemannian manifold $(\mathcal{M}, \mathfrak{g})$ is a smooth manifold $\mathcal{M}$ endowed with a Riemannian metric $\mathfrak{g}$. Each point on the manifold is associated with a tangent space where the metric $\mathfrak{g}$ is defined. The curvature is a notion describing the extent of how a manifold derivatives from being "flat". It is typically viewed as a measure $R(X, Y)Z$ of the extent to which the operator $(X, Y) \to \nabla_X\nabla_Y Z$ is symmetric, where $\nabla$ is a connection on $\mathcal{M}$ (where $X, Y, Z$ are vector fields, with $Z$ fixed). Sectional curvature, defined on two independent vector unit in the tangent space, is often utilized. The reason is the curvature operator $R$ can be recovered from the sectional curvature, when $\nabla$ is the canonical Levi-Civita connection induced by $\mathfrak{g}$. A manifold is said to be a Constant Curvature Space (CCS) if the sectional curvature is constant scalar everywhere on the manifold.

Among Riemannian manifolds, there exist three types of CCSs: the negative curvature hyperbolic space, the positive curvature hyperspherical space, and the zero-curvature Euclidean space. There are several model spaces of CCSs, e.g., Poincaré ball model, Poincaré half-plane, Klein model, Lorentz model, and Stereographical model, and they are equivalent to each other in essence[3]. In this paper, we opt for the Lorentz/Spherical model[4], and give a unified formalism,

$$\mathcal{L}^d_{\kappa} = \left\{ \begin{bmatrix} x_t \\ \boldsymbol{x}_s \end{bmatrix} \in \mathbb{R}^{d+1} | \langle\boldsymbol{x}, \boldsymbol{x}\rangle_{\kappa} = \frac{1}{\kappa}, x_t > 0, \boldsymbol{x}_s \in \mathbb{R}^d \right\}, \quad (22)$$

where $d$ and $\kappa$ denote the dimension and curvature, respectively. $x_t$ corresponds to the axis of symmetry of the hyperboloid and is termed the time-like dimension, while all other axes $\boldsymbol{x}_s$ are called space-like dimensions. In particular, $\mathcal{L}^d_{\kappa}$ becomes the Lorentz model of hyperbolic space under negative $\kappa$, and shifts to Spherical model of hyperspherical space when $\kappa > 0$. Note that, Euclidean space is not included in the formalism, and it requires $\kappa \neq 0$. The induced hyperbolic space is a $d$-dimensional upper hyperboloid embedded $(d + 1)$-dimensional Minkowski space, a.k.a. hyperboloid model. Similarly, the corresponding hyperspherical space is also expressed in a $(d + 1)$-dimensional space. All the mathematical construction in this paper is based on the Lorentz/Spherical model. Accordingly, given a point in the manifold $\boldsymbol{x} \in \mathcal{L}^d_{\kappa}$, the exponential map projects

---

**Algorithm 2:** Cross-geometry Attention in Hyperbolic Space

---

**Input:** A substructure, Node coordinates $\boldsymbol{p}^H$ and $\boldsymbol{p}^S$, Linear operation $f_W$, A parameterized scalar map $\phi : \mathcal{L} \times \mathcal{L} \to \mathbb{R}$.

**Output:** The updated node coordinates $\boldsymbol{p}^H$.

1 Compute the key, query and value via $\boldsymbol{k}_i = f_V(\boldsymbol{p}^H_i)$, $\boldsymbol{q}_i = f_Q(\boldsymbol{p}^S_i)$ and $\boldsymbol{v}_i = f_V(\boldsymbol{p}^H_i)$, respectively;

2 Compute the score of $\phi([\boldsymbol{q}_i, \boldsymbol{k}_j])$ for $i, j$ in the substructure;

3 Derive attentional weight by the softmax of scores over the substructure $\alpha_{ij} = \frac{\exp(\phi([\boldsymbol{q}_i||\boldsymbol{k}_j]))}{\sum_{(i,t)\in\Omega}\exp(\phi([\boldsymbol{q}_i||\boldsymbol{k}_t]))}$;

4 Update node coordinate by the weighted geometric midpoint $\boldsymbol{v}_i = mid_{\kappa}(\{\boldsymbol{v}_j, \alpha_{ij}\}_{(i,j)\in\Omega})$;

---

a vector $\boldsymbol{v}$ in the tangent space at $\boldsymbol{x}$ to the manifold $Exp_{\boldsymbol{x}}(\boldsymbol{v})$ : $\mathcal{T}_{\boldsymbol{x}}\mathcal{L}^d_{\kappa} \to \mathcal{L}^d_{\kappa}$, and the closed form expression is given as follows,

$$Exp_{\boldsymbol{x}}(\boldsymbol{v}) = \cos_{\kappa}(\sqrt{|\kappa|}\|\boldsymbol{v}\|_{\kappa})\boldsymbol{x} + \sin_{\kappa}(\sqrt{|\kappa|}\|\boldsymbol{v}\|_{\kappa})\frac{\boldsymbol{v}}{\sqrt{|\kappa|}\|\boldsymbol{v}\|_{\kappa}}. \quad (23)$$

The logarithmic map $Log_{\boldsymbol{x}}(\boldsymbol{y}) : \mathcal{L}^d_{\kappa} \to \mathcal{T}_{\boldsymbol{x}}\mathcal{L}^d_{\kappa}$ projects a point $\boldsymbol{y}$ in the manifold to the tangent space of $\boldsymbol{x}$, serving as the inverse of the exponential map. It takes the form of

$$Log_{\boldsymbol{x}}(\boldsymbol{y}) = \frac{\cos_{\kappa}^{-1}(-\kappa\langle\boldsymbol{x}, \boldsymbol{y}\rangle_{\kappa})}{\sqrt{\kappa^2\langle\boldsymbol{x}, \boldsymbol{y}\rangle^2_{\kappa} - 1}}(\boldsymbol{y} + \kappa\langle\boldsymbol{x}, \boldsymbol{y}\rangle_{\kappa}\boldsymbol{x}). \quad (24)$$

The parallel transport maps between two tangent spaces, and its closed form expression is given in Appendix B.

## E  Experiment Details

### E.1  Datasets

We give the statistics in Table 6, and introduce the datasets below.

- **Citeseer** [43] consists of scientific publications in six classes. Nodes and edges denote publications and citation relationship, respectively. Each publication is described as a binary word vector from the dictionary of 3703 unique words.
- **Pubmed** [43] is citation network among scientific publications in three classes. Each publication is described by a TF/IDF weighted word vector from a dictionary of 500 unique words.
- **GitHub** [31] is a social network where nodes are developers who have starred at least 10 repositories, and edges denote mutual follower relationships. Node features are location, starred repositories, employer, and e-mail address.
- **Airports** [30] is a commercial air transportation network within the United States. The node corresponds to a distinct airport facility, and are stratified into four discrete classes. The edges indicate the existence of commercial flight routes.
- **ogbn-arxiv** [15] is the citation network among Computer Science (CS) arXiv papers. Each paper is given as a 128-dimensional feature vector by averaging the embeddings of words in its title and abstract.
- **Physics** [33] is co-authorship graphs based on the Microsoft Academic Graph from the KDD Cup 2016 challenge. Nodes and edges denote authors and co-authored relationship, respectively.

---

[3]They are the same in structure and geometry but have different coordinate systems.
[4]The Lorentz model of hyperbolic space corresponds to the Spherical model of hyperspherical space in account of the coordinate systems.

**Table 6: Summary of Datasets**

| Dataset | #(Nodes) | #(Edges) | Feature Dim. |
|---|---|---|---|
| Cora | 2,708 | 5,429 | 1,433 |
| Pubmed | 19,717 | 44,338 | 500 |
| GitHub | 37,700 | 578,006 | 0 |
| Airports | 1,190 | 13,599 | 0 |
| ogbn-arxiv | 169,343 | 1,166,243 | 128 |
| Physics | 34,493 | 495,924 | 8,415 |
| AComp | 13,752 | 491,722 | 767 |

- **AComp** (Amazon Computers dataset) [33] is segments of the Amazon co-purchase graph. Nodes denote goods and edges indicate that two goods are frequently bought.

## E.2 Baselines

- **GCN** [20] resorts neighborhood aggregation in spectral domain.
- **DGI** [37] introduces a self-supervised paradigm by maximizing the mutual information between the local node view and the global graph view.
- **GraphMAE2** [14] conducts self-supervised learning in the reconstruction of masked node features with masked autoencoders.
- **OFA** [22] describes all nodes and edges with natural language to feed into LLMs, and subsequently utilizes graph prompting that appends prompting substructures to the input graph.
- **LLaGA** [5] re-organizes graph nodes to sequences and then maps the sequences into the token embedding space via a versatile projector in order to leverage the LLM for graph analysis.
- **OpenGraph** [40] is trained on diverse datasets with a unified graph tokenizer, scalable graph transformer, and LLM-enhanced data augmentation, so as to comprehend the nuances of diverse graphs.
- **GCOPE** [47] is a graph pre-training framework designed to enhance the efficacy of downstream tasks by harnessing collective insights from multiple source domains.
- **GraphAny** [48] models the inference on a new graph as an analytical solution to a GNN with designs invariant to feature and label permutations and robust to dimension changes.

## E.3 Reproducibility & Implementation Notes

*E.3.1 **On Few-shot Learning**.* Few-shot learning performance is significant to evaluate a pre-trained model. In particular, a pre-trained model is examined by classifying new data, which has not been seen during training, with only a few labeled samples for each class. In our experiment, following the setting of Xia et al. [40], we retain up to $k$ training instances for labeled classes. For example, we first pre-train our model on ogbn-Arxiv [15], Amazon Computers [33], and Coauthor Physics [33] datasets, and then fetch $k$ samples per class on Citeseer [43] to train the classification head, so as to infer the classification results.

*E.3.2 **Initialization and Configurations**.* For model initialization, we first compute the normalized graph Laplacian $L = I - D^{-1/2}AD^{-1/2}$ of the given graph, where $A$ is the adjacency matrix and $D$ is the degree matrix. Second, we conduct eigenvalue

**Table 7: Geometric ablation on Citeseer, Pubmed, and Airport datasets. Node classification results are reported in terms of AUC (%). The results are given in the form of mean±std. $\mathcal{R}_0^{32}$ denotes the Euclidean space.**

| Trees | Cycles | Citeseer | Pubmed | Airport |
|---|---|---|---|---|
| $\mathcal{H}_{-1}^{32}$ | $\mathcal{S}_1^{32}$ | **66.38 ± 0.31** | **76.20 ± 0.79** | **55.29 ± 2.26** |
| $\mathcal{H}_{-1}^{32}$ | $\mathcal{R}_0^{32}$ | 66.26 ± 1.45 | 73.10 ± 6.36 | 50.42 ± 1.48 |
| $\mathcal{H}_{-1}^{32}$ | $\mathcal{H}_{-1}^{32}$ | 63.37 ± 1.69 | 72.26 ± 2.12 | 52.66 ± 1.46 |
| $\mathcal{H}_{-1}^{32}$ | $\mathcal{S}_1^{32}$ | **66.38 ± 0.31** | **76.20 ± 0.79** | **55.29 ± 2.26** |
| $\mathcal{R}_0^{32}$ | $\mathcal{S}_1^{32}$ | 65.52 ± 1.46 | 71.12 ± 8.73 | 50.17 ± 1.26 |
| $\mathcal{S}_1^{32}$ | $\mathcal{S}_1^{32}$ | 64.26 ± 1.09 | 71.46 ± 0.72 | 53.72 ± 0.46 |

**Table 8: Cross-domain node classification prformance on different pre-training datasets.**

| Pre-training | Method | Testing Datasets | | |
|---|---|---|---|---|
| | | Citeseer | Pubmed | Airport |
| Flickr | OpenGraph | $63.16_{\pm4.45}$ | $60.35_{\pm5.53}$ | $43.32_{\pm2.23}$ |
| | GCOPE | $64.47_{\pm2.87}$ | $72.48_{\pm0.97}$ | $36.74_{\pm2.38}$ |
| | **RiemannGFM** | $\mathbf{65.20_{\pm1.73}}$ | $\mathbf{74.04_{\pm0.53}}$ | $\mathbf{46.13_{\pm2.78}}$ |
| AComp | OpenGraph | $60.24_{\pm1.25}$ | $64.45_{\pm1.24}$ | $45.02_{\pm4.25}$ |
| | GCOPE | $63.79_{\pm0.88}$ | $72.80_{\pm2.14}$ | $44.19_{\pm1.53}$ |
| | **RiemannGFM** | $\mathbf{64.80_{\pm1.96}}$ | $\mathbf{77.00_{\pm0.42}}$ | $\mathbf{49.41_{\pm1.77}}$ |
| WikiCS | OpenGraph | $\mathbf{67.54_{\pm2.24}}$ | $74.98_{\pm3.25}$ | $48.92_{\pm1.22}$ |
| | GCOPE | $65.47_{\pm2.87}$ | $75.38_{\pm0.83}$ | $46.05_{\pm2.51}$ |
| | **RiemannGFM** | $66.56_{\pm1.15}$ | $\mathbf{75.78_{\pm1.36}}$ | $\mathbf{51.25_{\pm1.76}}$ |

decomposition on $L$ and utilize the largest $K$ eignvectors as node encodings, where $K$ is a predefined number. Note that, the initialization process indeed normalizes different graphs with the $K$-dimensional encoding $z$. Subsequently, we induce node coordinates via $p = Exp_o([0||z^\top]^\top)$ so that the coordinates are placed on the manifold $p \in \mathcal{L}$, where the reference point is the north pole $o$. RiemannGFM allows for mini-batch training, and the mini-batch sampling strategy is the same as that of SAGE [13].

*E.3.3 **Hyperparameters**.* The hyperparameters are tuned with grid search. In particular, we set the dropout rate as 0.1 to enhance the model robustness, and set learning rate of the pre-training as 0.01 to balance convergence speed and stability. The dimension of each factor in the product bundle is set as 32, that is, we instantiate RiemannGFM on $\left(\mathcal{H}_{-1}^{32} \otimes \mathcal{TH}_{-1}^{32}\right) \otimes \left(\mathcal{S}_1^{32} \otimes \mathcal{TS}_1^{32}\right)$. RiemannGFM is implemented with 2 Riemannian layers. The parameterized scalar map in cross-geometry attention is a multi-layer perceptions with one hidden layer, whose dimension is set as 256. The model is built on PyTorch, and further details are provided in the anonymous link of https://anonymous.4open.science/r/Geo-GFM-1603.

## F Additional Results

Owing to the limit of space, we show the additional results of the geometric ablation in Table 7 and the impact of pre-training datasets in Table 8. The geometric ablation in node classification exhibits the similar pattern to that in link prediction, showing the alignment between trees and hyperbolic space (and between cycles and hyperspherical space). As shown in in Table 8, the stable performance of our model demonstrates the superiority of exploring GFM with structural vocabulary (i.e., the common substructures of trees and cycles underlying the graph domain).

