# OpenReview forum: "RiemannGFM: Learning a Graph Foundation Model from Structural Geometry"
_ACM.org/TheWebConf/2025/Conference — WWW 2025 Oral_

### Official Review · Reviewer_cmta · 2024-11-14

**Novelty:** 5
**Technical Quality:** 6

**Review:**

The paper proposes to develop a graph foundation model, RiemannGFM, that aims to capture and generalize structural knowledge across diverse graphs. Unlike existing models that primarily leverage text-attributed graphs or attempt to adapt language model techniques to graph data, RiemannGFM proposes a structural vocabulary built on fundamental graph substructures—specifically trees and cycles—mapped within a Riemannian geometric space. The key intuition is to integrate Riemannian geometry to allow for the learning of both local and global graph structures. Experiments on 4 benchmark datasets demonstrate the effectiveness of RiemannGFM in comparison with various baselines.

Strengths:
+ Building a graph foundation model that capture structural knowledge across diverse graphs is a crucial task in the graph mining community currently. The general idea of representing any graph via these foundational substructures is also well-motivated.

+ The proposed technique that uses Riemannian geometry to accommodate diverse graph types beyond text-attributed ones is novel from my perspective. RiemannGFM captures complex graph relationships within the manifold, enabling the model to maintain high levels of cross-domain transferability and generalization.

+ Experiments are comprehensive. 4 datasets from different domains are considered. Different types of baselines, including traditional GNNs, self-supervised models, and graph foundational models, are compared. Besides performance comparison, the authors also study the ablation version, the effect of pre-training datasets, and the expressiveness of structural knowledge to validate some of their design choices.

Weaknesses:
- Statistical significance tests are not conducted. It is not clear whether the gaps between RiemannGFM and the baselines/ablation versions are statistically significant or not in Tables 1-4. In fact, the improvement is subtle in some columns given the reported standard deviation.

- Only node classification and link prediction tasks are examined. Considering that the original motivation of building foundational models (e.g., in NLP) is to generalize the model to not only different domains but also different tasks, it would be beneficial to consider a wider range of tasks, such as node clustering, community detection, and node ranking.

**Questions:**

- Could you conduct a statistical significance test (e.g., two-tailed t-test) to compare RiemannGFM with the strongest baseline/ablation version in each table?

- Could you conduct experiments on more types of downstream tasks?

**Reviewer Confidence:**

3: The reviewer is confident but not certain that the evaluation is correct

**Scope:**

3: The work is somewhat relevant to the Web and to the track, and is of narrow interest to a sub-community

---

### Official Review · Reviewer_UWPf · 2024-11-22

**Novelty:** 5
**Technical Quality:** 4

**Review:**

The motivation of this paper is to address the limitations of existing graph models in terms of generalization ability and cross-domain transfer learning. The authors introduce methods from Riemannian geometry and align the shared substructures (i.e., structural vocabulary) present in graph domains. By leveraging the concept of product bundle in Riemannian geometry, the study unifies the universal structures of different types of graphs, thereby enhancing the model's adaptability and generalization capability in cross-domain transfer tasks.

Advantage 1: This approach innovatively combines hyperbolic space and spherical space to model structural lexicons, thereby overcoming the limitations of traditional Euclidean space and enabling the unified representation of general graph structures in higher-dimensional spaces.
Advantage 2: This method cleverly utilizes the concept of parallel transport from Riemannian geometry to compute the contrastive loss between the hyperbolic space view and the spherical space view, ensuring that different views of the same graph are more closely aligned in the representation space.

However, there are some issues that need to be improved or clarified, see **Questions** for details.

**Questions:**

Q1: The motivation and principles behind the use of Riemannian geometry in this paper remain unclear and need to be described and explained in further detail.

Q2: The authors' reasoning for associating tree structures with hyperbolic space, based solely on the characteristic that tree branches can extend infinitely without intersection, appears somewhat forced. Similarly, the connection between cyclic structures and spherical space is established merely due to their cyclic and symmetric properties. This reasoning seems rather tenuous and lacks sufficient theoretical justification. The authors should provide a more convincing logical explanation.

Q3: The description in the model section of this paper needs to be further optimized for organizational structure to improve readability.

Q4: The experiments in this paper lack the necessary convincing and fairness.

----(1) The experimental datasets in this paper are too few and unrepresentative. In addition to the **Citeseer and Pubmed** datasets .

----(2) The experimental performance results of the baseline methods on the comparable datasets differed significantly from the results of their original papers, which affected the reliability of the technical quality of this paper. (vanilla GNNs and GFM)

----(3) While this paper aims to address the limitations of existing graph models in terms of generalization ability and cross-domain transfer learning, the experimental results indicate that RiemannGFM performs suboptimally on the two datasets (**Citeseer and Pubmed**). The authors should provide a more thorough analysis and explanation to clarify the model's poor performance on these datasets.

----(4) The baselines for the vanilla GNNs group lack SOTA methods such as MaskGAE, S2GAE, etc.

**Reviewer Confidence:**

3: The reviewer is confident but not certain that the evaluation is correct

**Scope:**

4: The work is relevant to the Web and to the track, and is of broad interest to the community

---

### Official Review · Reviewer_ekjU · 2024-11-28

**Novelty:** 6
**Technical Quality:** 6

**Review:**

This paper introduces **RiemannGFM**, a novel graph foundation model that leverages Riemannian geometry and structural vocabulary to provide universal graph representation. The authors address key limitations in existing graph learning frameworks, particularly their dependence on textual attributes and Euclidean embeddings, by proposing a model that embeds structural elements—trees and cycles—into hyperbolic and hyperspherical spaces. The core contributions include a novel product bundle framework, a universal Riemannian layer with cross-geometry attention and bundle convolution, and a geometric contrastive learning method that ensures robust performance across diverse graph domains. Extensive experiments demonstrate the model's strong performance in cross-domain transfer, few-shot learning, and structural knowledge expressiveness.

**pros:**

- The paper introduces a unique approach to graph representation, combining Riemannian geometry with structural vocabulary to achieve universality.
- The authors provide detailed mathematical proofs and rigorous formulations for manifold-preserving operations, geometric midpoints, and attention mechanisms.
- RiemannGFM consistently outperforms or matches state-of-the-art models in cross-domain transfer learning and few-shot tasks across diverse datasets, showcasing its practical potential.

**cons:**

- My main concern lies in the scalability of the model, as the computational cost of its Riemannian operations warrants a more thorough analytical discussion to assess its practicality for large-scale graphs.

**Questions:**

See above cons.

**Reviewer Confidence:**

2: The reviewer is willing to defend the evaluation, but it is likely that the reviewer did not understand parts of the paper

**Scope:**

4: The work is relevant to the Web and to the track, and is of broad interest to the community

---

### Official Review · Reviewer_gpnd · 2024-12-01

**Novelty:** 6
**Technical Quality:** 6

**Review:**

Pros:
1. There are solid mathematical proofs in your paper.
2. Strutural properties are of great concern in GFM and your work is insightful in such field.
3. Extensive experiments, including node classification, few-shot learning and geometric ablation, have testified the effectiveness of your model.

Cons:
1. There lacks visualization on the difference between your model and other GFMs. For example, you can visualize the embeddings under fair conditions.
2. In Table 1, your model does not work well on Citeseer and Pubmed in node classification while it outperforms baselines in link prediction. I think you should give further explanation on special cases as well as how your model works well in link prediction.

**Questions:**

1. Can your model be transplanted to graph-level tasks such as graph-level classification/clustering? I think that graph-level tasks are tougher than node-level tasks and such tasks are in great need of GFMs.
2. How to use  Riemann geometry on other kinds of structural properties, such as cliques and centrality distributions?

**Reviewer Confidence:**

3: The reviewer is confident but not certain that the evaluation is correct

**Scope:**

4: The work is relevant to the Web and to the track, and is of broad interest to the community

---

### Official Review · Reviewer_NwrB · 2024-12-02

**Novelty:** 5
**Technical Quality:** 5

**Review:**

This paper proposes learning 'graph vocabulary' for graph foundation models using Riemannian geometry. The idea is to represent structural information in tree and loop structures, encode the structures using a geometric attention layer, and use contrastive learning for pre-training.

Strength:
1. The graph foundation model is an important and novel research direction. The proposed methods are reasonable and novel.
2. The proposed graph vocabulary using tree and loop structures, and the encoder inspired by Riemannian geometry are rigorously proved and may lead to many potential future works.
3. The experiment section is comprehensive.

Weakness:
1. I'm not familiar with 'Riemannian geometry'. Many paragraphs and equations are difficult for me to understand. The parameter labels are often complicated with multiple layers of superscripts or subscripts. For example, in line 189, the tangent space T_xM is first defined, but it's unclear what the operators and relations are between 'T,' 'x,' and 'M.' In line 344, eq.(2) is complicated; some clarifications on what each term means can be very helpful. In summary, I recommend the author clarify the parameters and add some explanations to each subsection and equation to explain the core idea in simple words.
2. From what I understand, eq.(5)(6) states the structural vocabulary is encoded using a weighted mean with attention. I feel this design is overly simplified by encoding a subgraph into the mean of node features.
3. In line 462 eq.(7), the author proposes 'Bundle Convolution for message passing over tangent bundles' to replace the regular message-passing GNN. This section needs further clarification. What does each term in eq.(7) mean? Why can it solve the incompatible problem? How is the proposed formulation related to the regular GNN formulation?

**Questions:**

Q1. The main challenge of the graph foundation model is the subject diversity (for example, biology vs social networks). For example, the molecule graphs and citation graphs are very different in terms of both node attributes and structural patterns. How can your proposed method solve this diversity? Are tree and loop structures enough?

**Reviewer Confidence:**

3: The reviewer is confident but not certain that the evaluation is correct

**Scope:**

3: The work is somewhat relevant to the Web and to the track, and is of narrow interest to a sub-community